# Epidemiological Survey and Retrospective Analysis of *Salmonella* Infections between 2000 and 2017 in Warmia and Masuria Voivodship in Poland

**DOI:** 10.3390/medicina55030074

**Published:** 2019-03-20

**Authors:** Paula Dmochowska, Maria Spyczak von Brzezinski, Jacek Żelazowski, Joanna Wojtkiewicz, Susanne Jung, Joanna M. Harazny

**Affiliations:** 1Department of Pathophysiology, University of Warmia and Masuria, PL 10-082 Olsztyn, Poland; paula.dmochowska@gmail.com (P.D.); maria.svonb@gmail.com (M.S.v.B.); jacek.zelazowski@gmail.com (J.Z.); joanna.wojtkiewicz@uwm.edu.pl (J.W.); 2Clinical Research Center, Department of Medicine 4 Nephrology and Hypertension, Erlangen-Nuremberg University, D 91054 Erlangen, Germany; Susanne.Jung@uk-erlangen.de

**Keywords:** *Salmonella*, infectious diseases, Warmia and Masuria region, Poland, EU

## Abstract

*Background and Objectives*: Salmonellosis is a major foodborne bacterial infection throughout the world. Epidemiological surveillance is one of the key factors to reduce the number of infections caused by this pathogen in both humans and animals. The first outcome measure was the prevalence of non-typhoid *Salmonella* (NTS) infections between 2000 and 2017 among the population of the predominantly agricultural and touristic Polish region of Warmia and Masuria (WaM). The second outcome measure was the comparison of the NTS hospitalization rate of all registered NTS cases, an investigation of the monthly reports of infections, and the exploration of the annual minimal and maximal NTS infection number in WaM in the above-mentioned time period. The last outcome was a comparison of the prevalence of NTS infections in the region and in its administrative districts by considering both rural and urban municipalities three years before and three years after the accession of Poland into the European Union (EU) in 2004. Materials and *Methods:* The total number of infections and hospitalizations in the 19 districts of the WaM voivodship in Poland was registered monthly between 2000–2017 by the Provincial Sanitary-Epidemiological Station in Olsztyn, Poland. *Results:* Between 2000 and 2017, the number of diagnosed salmonellosis cases decreased significantly in WaM; the decrease was higher in urban districts than in rural ones, and the ratio of hospitalizations and the total number of NTS cases increased significantly across all districts. The lowest number of cases was reported in the winter months and was stable from 2007, whereas the highest number was reported in the summer months with a higher tendency of outbreaks. *Conclusion*: The falling number of salmonellosis cases in 2000–2017 in WaM reflects the general trend in Poland and Europe. The decrease of NTS infections in WaM is related to the accession of Poland into the EU.

## 1. Introduction

Salmonellosis is a major foodborne bacterial gastroenteritis infection throughout the world. According to the World Health Organization (WHO), the main source of infection is from animal source foods, such as meat, poultry, eggs, and dairy products. Common transmission routes are fecal–oral from person to person or contaminated food from catering services. The other sources are pets, including cats, dogs, birds, and reptiles such as turtles, as well as green vegetables contaminated by manure [1]. In 2013, Hernandez-Reyes and Schikora described the transmission of salmonellosis via insects to fruits, nuts, lettuce, tomatoes, sprouts, sesame seeds, vine vegetables, and roots [2].

Common symptoms of salmonellosis are abdominal cramps, diarrhea, fever, headache, and vomiting, which develop 12–72 h after the infection and usually last between four and seven days [3]. Mostly, infections caused by non-typhoid *Salmonella* (NTS) are self-limiting. Nevertheless, in patients with immunodeficiency, the elderly, children, or patients with co-infections, the infection may be fatal, especially for people from Africa, where human immunodeficiency virus (HIV) infection is over 70% of the global burden and where the population has a high risk of mortality by NTS [4,5].

The main indications for hospitalization are exhaustion and dehydration [6]. Non-typhoid salmonellosis is a major cause of diarrheal diseases worldwide and causes approximately 93 million enteric infections and 155,000 deaths annually [7]. Food can be contaminated at every stage of production. Even well-developed countries have failed to eliminate NTS bacteria completely. Painter et al. described the attribution of foodborne infection, hospitalization, and death to food commodities in the USA [8] between 1998 and 2008 and estimated that, in this timeframe, over 3.5 million foodborne infections were caused by more than 20 different bacterial infections. More deaths were attributed to poultry products (19%) than to any other commodity, and most poultry-associated deaths were caused by *Listeria* or *Salmonella* spp. In neighboring Canada, salmonellosis dominated as a cause of hospitalization among other domestically acquired foodborne infections [9].

*Salmonella* serovars, enteritidis, and typhimurium cause gastroenteritis in low-income countries as well as in industrialized countries [10]. In 2006, Majowicz et al. estimated about 690 cases of salmonellosis per 100,000 inhabitants in Europe [11].

Between December 2014 and April 2015, there was an epidemic outbreak of *Salmonella* in France, which was linked to the consumption of beef burgers contaminated with non-typhoid *Salmonella* originating from a Polish producer. Forty-one people in northern France were infected after the consumption of the contaminated meat [12]. This incident triggered a demand for information on the prevalence of NTS infections in the population of the predominantly agricultural Polish voivodship of Warmia and Masuria (WaM, Polish: *województwo warmińsko-mazurskie*). WaM is one of 16 administrative subdivisions—voivodships—in northeastern Poland and is an important food producer. The WaM region consists of an area of 24,192 km^2^ with 1,439,675 inhabitants (3.75% of the Polish population which had 38,437,239 inhabitants in 2017). Olsztyn is the capital and largest city in the region. The population density in WaM was 59.5 inhabitants/km^2^ with 59.33% of urbanization vs. 123 inhabitants/km^2^ and 60.54% urbanization for the whole of Poland in 2017. In 2017, the demographic data of the European Union showed 505,701,172 inhabitants with 115 inhabitants/km^2^ and 74.58% of urbanization [13,14]. Administratively, the WaM voivodship is divided into 19 districts: Bartoszyce, Braniewo, Działdowo, Elbląg, Ełk, Giżycko, Gołdap, Iława, Kętrzyn, Lidzbark, Mrągowo, Nidzica, Nowe Miasto, Olecko, Olsztyn, Ostróda, Pisz, Węgorzewo, and Szczytno.

The Great Masurian Lakes region is especially attractive for tourists. Between 2015 and 2016, the number of rooms rented by tourists increased by 10%. Outbreaks of NTS infections caused by food imported from Poland could be economically relevant for WaM, one of the most underprivileged voivodships in terms of economic status in Poland. In 2016, 5.09% inhabitants of WaM were registered as unemployed, compared with a mean of 3.47% for the whole of Poland [15,16,17].

In 2004, Poland joined the European Union (EU), which determined the implementation of food production guidelines required by EU law (General Regulations regarding the control of *Salmonella*-act EC No. 2160/2003). A procedure for the monitoring of zoonoses and zoonotic agents was developed in 2003 by establishing the European Food Safety Authority (EFSA) in 2002. The EFSA determines the general principles and requirements of food laws as well as procedures in matters of food safety monitoring, data collection, and analysis in order to identify emerging risks concerning food safety, in accordance with the provisions in Regulation (EC) No. l78/2002 3 based on Directive 2003/99/EC of the European Parliament and the Council of 17 November 2003 on the monitoring of zoonoses and zoonotic agents. The process of Poland’s integration into the EU began with Poland’s application for membership in Athens on 8 April 1994.

The primary aim of this work was the collection of baseline data and the evaluation of the prevalence of NTS infections, hospitalizations, and expenditure of NTS therapy in WaM between 2000 and 2017. The second aim was to investigate the monthly reports of the infections and to check the annual minimal and maximal NTS infections number between 2000 and 2017. Another point of interest was to compare the prevalence of NTS infections in the region and in its administrative districts with consideration of rural and urban municipalities before (2000–2002) and after (2015–2017) Poland’s accession into the EU in 2004.

## 2. Materials and Methods

In the event of a suspected or diagnosed *Salmonella* infection, the physician is obliged to report the case to the sanitary inspector of the district within 24 h, according to Polish law regarding infectious diseases and infections (Dz.U. 2018 poz. 151 on 18 January 2018). Every two weeks, the sanitary inspector has to report the data to the Provincial Sanitary-Epidemiological Station (WSSE) located in the capital of the respective voivodship. The data are reported annually to the Chief Sanitary Inspector, who is the central body of government administration, subordinate to the Ministry of Health in Poland since 1 January 2000

For the study, the monthly data of the prevalence and hospitalization of NTS infections between 2000 and 2017 from all the districts of WaM were extracted from the WSSE database in Olsztyn with the agreement of the Head of the WSSE. Our research data were restricted to NTS infections. We extracted all available data from the database including month, year, location, and information about ambulatory treatment or hospitalization for every patient with a NTS infection in WaM. Data regarding the gender and age of the patients or *Salmonella* strain causing the NTS contamination in individuals were not available in the WSSE database.

We also analyzed the average expenditure of *Salmonella* infection therapy in WaM between 2000 and 2017. The current average costs of *Salmonella* infection treatment in Poland are 67.00 PLN (16 EUR) for ambulatory care and 2480.00 PLN (575 EUR) for hospital treatment per capita [18].

The data were statistically analyzed and illustrated by IBM^®^ SPSS^®^ Statistics version 23 (IBM Corp., New York, NY, USA). The Figure 4 was draw by CorrelDraw^®^ Graphic Suite 2018PL (CorrelDraw^®^, Ottawa, Canada). None of the explored parameters were normally distributed, as confirmed by the Kolmogorov–Smirnov test. Therefore, for correlation analysis between parameters and a comparative analysis of the means/medians, non-parametric tests were used (the Mann–Whitney U test for unpaired median values, the Wilcoxon test for paired median values, and Spearman’s correlation). A *p* value < 0.05 was considered statistically significant.

## 3. Results

The number of NTS cases in both WaM and Poland decreased significantly (*r* = −0.922, *p* < 0.001 and *r* = −0.812, *p* < 0.001, respectively, based on Spearman’s correlation) during the studied time period. In Poland from 2008, stabilization followed the continuous decrease of NTS cases (Figure 1).

In 2000, the number of diagnosed NTS infections per 100,000 people was higher in WaM by 29.9 per 100,000 inhabitants than across the whole of Poland (50.4%). This difference has decreased, particularly since 2005 (the first whole year after Poland’s accession into the EU), and, in 2017, the number of NTS cases per 100,000 people was lower than the mean number of NTS infections in Poland, by 2.84 per 100,000 inhabitants (–10.9%).

In 2000, 1260 cases of NTS infections were recorded in WaM while, in 2017, only 333 NTS cases were assessed, i.e., a decrease of 73.6%. The highest annual decline of the analyzed period was in 2005 (Figure 1).

During the study period, we observed an increase in the rate of hospitalizations for reported NTS patients in WaM. In 2000, 55% of the infected NTS patients were hospitalized in WaM, compared with 71% in 2017. In Poland, 65% of NTS patients were hospitalized in 2000, compared with 63% in 2017.

The total expenditure of NTS treatment in WaM decreased significantly, by 66% (*r* = −0.92, *p* < 0.001 based on Spearman’s correlation), and permanently as the number of NTS cases fell.

We observed a seasonal pattern with a marked peak of NTS incidents in summer and the minimum NTS incidence in winter. As the general number of NTS cases decreased, the summer peak became less emphasized with a decrease over time (Figure 2). 

The minimum number of monthly NTS cases each year decreased significantly (*r* = −0.726, *p* = 0.001 based on Spearman’s correlation) throughout the studied time period and has been stable since 2007, with approximately 10 NTS cases per 10,000 inhabitants.

There was a significant decrease in the maximum number of monthly NTS cases per year throughout the studied time period (*r* = −0.804, *p* < 0.001 based on Spearman’s correlation) and, from 2008 onward, there was a recognizable tendency toward stabilization. Nevertheless, there were noticeable outbreaks in 2011 and 2013 (Figure 3).

In order to explore the effect of Poland’s accession into the EU on the number of diagnosed NTS infections, the first three years of the studied time period before accession (2000–2002) and the last three years (2015–2017) were compared separately for every district of WaM (Figure 4). The period between 2015 and 2017 represents the most recent years of the studied period after Poland’s accession into the EU. With 4.65 cases per 10,000 inhabitants, the mean number of reported NTS cases of the district of Elbląg was the lowest in the years 2000–2002. The highest number was registered in the districts of Bartoszyce, Giżycko, Gołdap, Olecko, and Węgorzewo, with more than 12.1 NTS cases per 10,000 inhabitants. In the years 2015–2017, the lowest mean number of NTS cases was observed in the district of Lidzbark, with 0.87 cases per 10,000 inhabitants, and the highest NTS number was registered in Działdowo, with 3.7 cases per 10,000 inhabitants. A comparison of the first and last three years of the studied time period indicated a significant decrease (*p* < 0.001) in NTS cases in each of the districts. In WaM, the mean NTS prevalence and the number of hospitalized NTS cases per 10,000 inhabitants in the years 2015–2017 decreased significantly by 68 ± 17% and 75 ± 13%, respectively, compared to the three years before Poland’s accession into the EU in 2004 (Table 1). We observed both a decrease in the prevalence of NTS and a 14.3% increase in mean rate of NTS hospitalization relative to the prevalence in 2015–2017 in comparison to 2000–2002 in WaM.

In 2000–2002, the mean number of reported NTS cases per 10,000 inhabitants was similar in both the rural and urban districts when compared to the mean values of 2015–2017. A significant difference of 35% between the rural and urban districts was found in the mean values of the hospitalized NTS cases per 10,000 inhabitants in 2015–2017. Before Poland’s accession into the EU, the difference was 11% and was not significant (Table 2). The decrease in NTS infection prevalence per 10,000 inhabitants was significantly higher in the urban districts, i.e., −82.9% (*p* = 0.003 by the Mann–Whitney U test), than in the rural districts, i.e., −74.1% (*p* = 0.018 based on the Mann–Whitney U test) in WaM.

## 4. Discussion

The incidence of *Salmonella* infections in WaM fell significantly between 2000 and 2017. The greatest decline was recorded in 2005, which may be related to the implementation of food production guidelines required by EU law after Poland’s accession into the EU in 2004. The EFSA program has been successful in all 28 EU countries, and the results have been described in detail for the United Kingdom [19]. The number of human infections decreased from more than 200,000 reported NTS cases per annum before 2004 and fewer than 90,000 NTS cases in 2014 across the EU [20].

Before Poland’s accession into the EU, the number of registered NTS infections per 100,000 inhabitants in WaM was higher (by 50.4%) than across all of Poland, but the difference decreased between 2000 and 2017. The number of NTS infections per 100,000 in WaM from 2015 onward was lower in WaM than in Poland, by 4.5% in 2015, 26.9% in 2016, and 11.5% in 2017 (Figure 1). We assumed that this was due to an improvement in food production and hygiene standards after Poland’s accession into the EU. Certainly, after 2004, the quality of diagnostics and therapy improved in the Polish healthcare system. Nevertheless, the reason for the increase in the ratio of NTS hospitalization to NTS prevalence, according to the developments in the healthcare system, remains unclear because the NTS hospitalization recommendations have not changed. During the last three years, WaM has developed into a tourist region, which has led to positive developments regarding the economy and living standards in the region, but this has also made the region susceptible to the risk of salmonellosis. 

In the rural and urban municipalities of WaM, we observed a discreete increase in the significances in the prevalence and hospitalization differences before (2000–2002) and after (2015–2017) Poland’s accession (Table 2). It is likely that in rural regions, direct work with NTS sources increases the risk of NTS contamination. The decrease in the differences between the prevalence before (2000–2002) and after (2015–2017) accession into the EU was higher in urban municipalities than in rural ones. The decrease in hospitalization before (2000–2002) and after (2015–2017) Poland’s accession was similar in both urban and rural municipalities. The lower hospitalization rate in urban municipalities, even significantly lower after Poland’s EU accession, suggests a milder course of the disease or faster access to health services than in rural municipalities. 

Furthermore, the rate of hospitalization relative to the prevalence of diagnosed salmonellosis increased in WaM during this time. According to unpublished observations from local experts of infectious diseases, the recommendations for NTS hospitalization did not change between 2000 and 2017. In this time, access to health care and NTS therapy in cases of NTS infection did not change noticeably in WaM. Local experts excluded the impact of health care changes since 2000 in the decrease of NTS prevalence in WaM between 2000 and 2017. However, the increase in the NTS hospitalization rate in WaM could be due to the lack of physician reports of ambulatory-diagnosed NTS infections because of a mild course of the disease. The high rate of reported hospitalizations of NTS cases in Poland was seen as an under-diagnosed and under-reported problem of NTS incidence [21]. In 2014, the rate of hospitalized cases in Cyprus, Greece, and Portugal was similar to that in WaM (75–85% vs. 71%, respectively) [22]. Mellou et al. described the under-reporting of NTS as a reason for an increase in the hospitalization rate in Greece [23]. Despite the fact that in the final account the percentage of hospitalizations increased, the total expenditure of salmonellosis treatment decreased significantly by two-thirds in WaM, which nowadays is a saving of approximately 300,000 Euro per year. The significant decrease in NTS cases after Poland’s EU accession was observed in all of the studied districts in WaM (Figure 4). Our results were in accordance with results from Greece, where, like WaM, the number of NTS cases has fallen in recent years. Mellou et al. found that, in 2012, the number of NTS cases in Greece was 3.6 per 100,000 inhabitants compared to 12 per 100,000 in 2004. In 2014, the incidence of salmonellosis in Greece was similar to that in other European countries, higher than other industrialized countries of the world but lower than that in Poland [13]. According to our study, the incidence of salmonellosis also decreased significantly in Poland, as in WaM, and was comparable to the economically stronger Germany in 2016 [24] (Poland’s neighboring country to the west, with similar culinary habits and geo-climate, but with approximately double the inhabitant density per km^2^ (232/km^2^ vs. 123/km^2^ in Poland [25])). A related decrease in the reported NTS cases was observed in Poland’s neighboring countries, which joined the EU at the same time (Latvia, Lithuania, and Estonia). In the EU, the number of reported NTS cases was 20.4 per 100,000 inhabitants and has been stable since 2012 [26]. 

With Poland’s accession into the EU, the gastronomic offerings in WaM have continuously changed, with more and more meals having a risk of NTS contamination (e.g., raw fish (sushi) and raw meat (carpaccio)) being offered. More innovative forms of meals, such as buffets or “dinner-aperitifs”, are tending to replace traditional meals [27]. During the winter months, the number of NTS cases in WaM has stabilized since 2007 at a level of 5–15 cases per 100,000 people per month; however, in summer, from 2008 onward, the number was about six times higher than that in winter with outbreaks of the disease in 2011 and 2013 (Figure 4). NTS infections by Polish food in France in 2014 occurred in seasons where higher temperatures favor the spread of bacterial diseases [12,28,29,30,31,32]. An increase in gastrointestinal bacterial infections in months with higher temperatures was observed globally, in Poland’s immediate neighbor, Latvia, as well as in countries outside of the EU, namely in the USA, Canada, and Australia [10,33,34,35]. However, Poland is an important beef producer. With 501,000 tons in 2016, Poland ranked seventh in the EU [26], and the uncontrolled NTS infections by Polish food led to economic depreciation. Considering this, the supervision of EU standards and improvement in the hygiene and quality of food processing, especially in summer, are of great importance. The reduction in NTS cases in WaM could have been caused by a combination of statutory requirements, as well as the need to obtain financial resources after moving away from the state economy. Fear of losing resources or high penalties for non-compliance in this area may be a good motivator to comply with the statutory recommendations.

## 5. Limitations

Ruzante et al. reported that outbreaks of different *Salmonella* types in Canada between 2003 and 2009 were associated with the seasons. Some *Salmonella* strains that appeared in winter were probably linked to international travel, whereas in summer the strains of bacteria were associated with domestically acquired infections [36]. In our research, there were no data available to obtain detailed information about the types of *Salmonella* strains. Data were not reported to the Provincial Sanitary-Epidemiological Station in Olsztyn by physicians, because reporting was not mandatory. We analyzed the data from the WSSE database, but unfortunately there were no data about the clinical conditions of the patients, diagnostic methods, or *Salmonella* strain available in the WSSE database. However, unpublished data from regional specialists of infectious diseases reported that *Salmonella* enteritidis was the most common pathogen responsible for NTS cases in WaM during that time.

The data did not reveal if the outbreak originated from one source and affected many people or if there were many sources affecting individuals. Unfortunately, there were also no data on patients regarding demographic groups, comorbidities, or immunosuppression.

## 6. Conclusions

The incidence of non-typhoid salmonellosis in WaM has decreased significantly during the last 18 years. With respect to *Salmonella* infections, the implementation of EFSA guidelines in Poland after accession into the EU in 2004 could have improved the health situation in WaM. The number of NTS cases in 2017 was 22 per 100,000 inhabitants and reached the average European level [22]. This resulted in a significant drop in the expenditure of *Salmonella* therapy, both in the WaM region as well as in all of its districts. Unlike during the winter months, there are still observed outbreaks of the disease in summer, suggesting that there is a need for the stronger control of food production in summer.

## Figures and Tables

**Figure 1 medicina-55-00074-f001:**
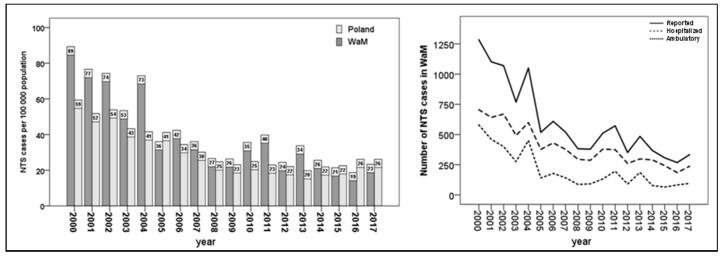
Non-typhoid *Salmonella* (NTS) cases per 100,000 inhabitants in WaM and Poland, and total NTS cases (reported, hospitalized, and ambulatory) in WaM from 2000 to 2017.

**Figure 2 medicina-55-00074-f002:**
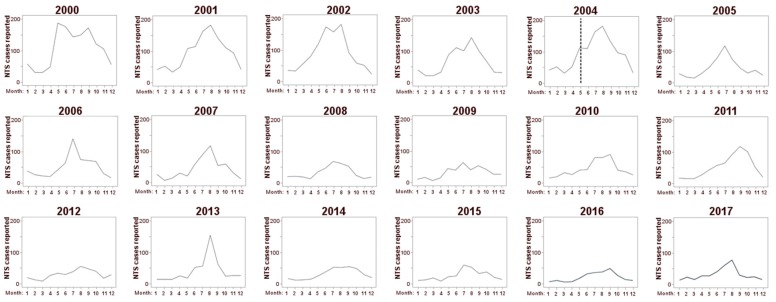
Number of monthly reported NTS cases in the studied years (2000–2017). The dotted line in 2004 marks the accession of Poland into the EU.

**Figure 3 medicina-55-00074-f003:**
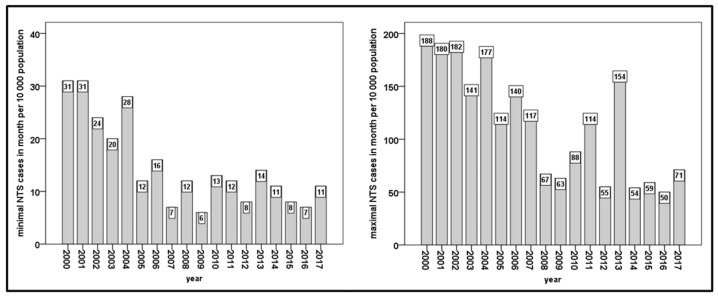
Minimum number of NTS cases (winter) and maximum number of NTS cases (summer) in WaM based on monthly reports from 2000 to 2017.

**Figure 4 medicina-55-00074-f004:**
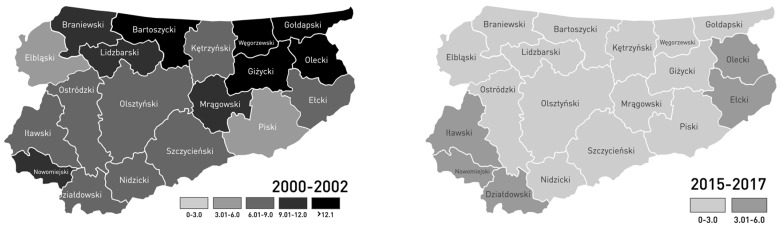
Comparison of the mean NTS cases per 10,000 inhabitants in the 19 districts of WaM before (averaged from 2000–2002) and after (averaged from 2015–2017) Poland’s accession into the EU.

**Table 1 medicina-55-00074-t001:** Comparison of the prevalence of NTS and hospitalized NTS cases per 10,000 inhabitants in the three years before Poland’s accession in 2004 into the European Union (before the EU in 2000–2002) with the last three years (after joining the EU in 2015–2017) in WaM.

		NTS Cases Per 10,000 Inhabitants			
	Before EU Median	Before EU IQR	After EU Median	After EU IQR	*p*
Prevalence	9.51	7.02; 12.34	2.10	1.56; 3.23	<0.001
Hospitalized	5.86	4.77; 6.72	1.66	1.00; 2.53	<0.001

Note: The *p* value was calculated by the non-parametric Wilcoxon test for paired median values.

**Table 2 medicina-55-00074-t002:** Comparison of the medians and interquartile range (IQR), between the 25th and 75th percentile, of the NTS prevalence and the number of hospitalized NTS cases per 10,000 inhabitants in the three years before Poland’s accession in 2004 into the European Union (between 2000 and 2002) with the last three years (after joining the EU, between 2015 and 2017) in the rural (rural municipalities ≥ 50%) and urban (rural municipalities < 50%) districts of WaM.

	Rural Municipalities N = 11		Urban Municipalities N = 7		*p*
	Median	IQR	Median	IQR	
Prevalence before EU	8.87	6.81; 9.65	10.22	7.09; 12.91	0.54
Prevalence after EU	2.30	1.85; 3.47	1.75	1.15; 3.05	0.13
Hospitalized before EU	6.23	5.72; 6.99	5.23	4.17; 6.63	0.29
Hospitalized after EU	1.77	1.62; 3.18	1.23	0.86; 2.28	0.044

Note: The *p* value was calculated by the non-parametric Mann–Whitney U test for the unpaired median values.

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
