# Peer review of "Epidemiological Survey and Retrospective Analysis of Salmonella Infections between 2000 and 2017 in Warmia and Masuria Voivodship in Poland"

_medicina, 2019, doi:10.3390/medicina55030074_

Round 1

Reviewer 1 Report

The reviewed paper is a research paper presenting the results of Salmonella surveillance in a polish region during the period 200-2017; it emphasizes a shift during this period probably linked to the application of European regulations in Poland that accessed to EU in that time.

The paper is interesting, and should be considered for publication. But it should be first read by a native English: the paper must be rewritten by correcting mistakes and making sure that certain sentences are no longer confusing. These confusing sentences should be rephrased.

One other major question resides in the methods used: can the authors make sure that the chosen statistical methods were appropriate?

Otherwise, a few minor remarks can also be made afterwards:

Line 16: please rephrase the confusing sentence

Line 23: voivodship means region and could be included in the title?

Line 24: the sentence “the data were evaluated” appears unnecessary and could be removed.

Lines 35-39: this sentence is too long and could be shortened. Please add the ref to WHO recommendations.

Lines 44-47: please rephrase the confusing sentence

Line 51: why Canada? Reason for the choice of this example?

Line 54 and elsewhere in the text (for instance lines 101, 110, 111…): “Salmonella” should be italicised

Line 62 voivodhip could be presented as the polish designation of an administrative region

Line 62: “northeaster” should “northeastern”

Lines 66-72: I do not understand, please rephrase the confusing sentence

Line 81: please add “of after “integration”

Lines 91-99: this paragraph should be included in the introduction, this is not M&M from my point of view

Line 100: why “LAW” in capital letters?

Line 101: “contracting” should read “contraction”?

Lines 100-103: please rephrase the confusing sentence; also, the law is from 2001 while the retrospective study starts in 200: is it OK?

Line 117: which tests? Please argue the choice; beware also of the change in the font

Lines 121-122, please rephrase the confusing sentence, as the stabilisation followed the decrease (not: “the decrease has stabilized”…)

Lines 155-157: please rephrase the confusing and too long sentence

Lines 177-183: please reconsider the paragraph that is confusing. Also are you sure that the results were significant as the IRQ overlap?? Please argue on that point.

Line 192: “in 2000-2017” should read “between 2000 and 2017”

Lines 201-202 please remove “as a consequence”

Lines 211-218: please rephrase the confusing sentence

Line 241 please add “meals” after “traditional”

Line 249: “was on the place 7” should read “ranked”, and please add “Poland before “ranked”

Author Response

Dear Reviewer,

We would like to thank you for all remarks and very accurate indications. We implemented them all and sent our manuscript to a native speaker to check our language. If you feel that our language is not sufficient, we are ready to send it to english editing office from Medicina

We are waiting for your opinion. 

Best regards,

Paula Dmochowska

Otherwise, a few minor remarks can also be made afterwards:

Line 16: please rephrase the confusing sentence

We rephrased this sentence. 

Line 23: voivodship means region and could be included in the title?

We changed the title.

Line 24: the sentence “the data were evaluated” appears unnecessary and could be removed.

We removed this sentence.

Lines 35-39: this sentence is too long and could be shortened. Please add the ref to WHO recommendations.

We shortened the list and added the ref but there are so many listed sources in detail according to the other recommendation.

 Lines 44-47: please rephrase the confusing sentence

We rephrased this sentence and shortened it.

Line 51: why Canada? Reason for the choice of this example?

·      We chose the Canada as a good example of a developed country to indicate that it is hard to find a place worldwide which defeated the problem of NTS infections. 

Line 54 and elsewhere in the text (for instance lines 101, 110, 111…): “Salmonella” should be italicized

In the text we italicized a word Salmonella.

Line 62 voivodhip could be presented as the polish designation of an administrative region

We changed it.

Line 62: “northeaster” should “northeastern”

We changed it.

Lines 66-72: I do not understand, please rephrase the confusing sentence

We changed it.

Line 81: please add “of after “integration”

We changed it.

Lines 91-99: this paragraph should be included in the introduction, this is not M&M from my point of view

We included this paragraph in the introduction.

Line 100: why “LAW” in capital letters?

We changed it.

Line 101: “contracting” should read “contraction”?

Yes, We changed it.

Lines 100-103: please rephrase the confusing sentence; also, the law is from 2001 while the retrospective study starts in 200: is it OK?

Thank you for your indication. We tried to solve it in the paper.

Line 117: which tests? Please argue the choice; beware also of the change in the font

Lines 121-122, please rephrase the confusing sentence, as the stabilisation followed the decrease (not: “the decrease has stabilized”…)

We corrected that sentence.

Lines 155-157: please rephrase the confusing and too long sentence

We corrected that sentence and shortened it.

Lines 177-183: please reconsider the paragraph that is confusing. Also are you sure that the results were significant as the IRQ overlap?? Please argue on that point.

It is answered on the text.

Line 192: “in 2000-2017” should read “between 2000 and 2017”

We corrected that sentence.

Lines 201-202 please remove “as a consequence”

We corrected that sentence.

Lines 211-218: please rephrase the confusing sentence

We rephrased this sentence.

Line 241 please add “meals” after “traditional”

We corrected that sentence.

Line 249: “was on the place 7” should read “ranked”, and please add “Poland before “ranked”

We corrected that sentence.

Reviewer 2 Report

In the article "Epidemiological Survey and Retrospective Analysis of Salmonella Infections in 2000-2017 in Warmia and Masuria Region in Poland" authors presented data for epidemiology of Salmonella infections in two regions of Poland. Overall, the manuscript needs editing for language and style. For many instances it was hard to understand what message authors wanted to convey. 

Please rearrange (re-write) the introduction to reflect what is already known in the field, knowledge gap and the approaches taken to answer the questions. Also, the methodology is insufficient. Please describe when and how these data were collected, analysis methods and interpretation. Why did the author only considered the non-typhoidal Salmonella infection? What about other Salmonella infections that cause enteric fever? How about clinical conditions of these patients? How did these patients were diagnosed? Are these patients culture confirmed? What other changes occur, other than implementation of food processing guidelines following accession to the EU, which made the dramatic reduction in NTS cases in these regions?   

Author Response

Dear Reviewer,

We would like to thank you for all remarks and very accurate indications. We implemented them all and sent our manuscript to a native speaker to check our language. If you feel that our language is not sufficient, we are ready to send it to english editing office from Medicina. 

We are waiting for your opinion. 

We rearranged the introduction as you asked us to do it. We improved the methodology by adding information and describing it in limitations. We answered for all of your questions you indicated.

Best regards,

Paula Dmochowska

Round 2

Reviewer 1 Report

Revision Medicina paper Salmonella in Poland

The paper is still interesting, and should be considered for publication. Some confusing sentences have been rephrased and are no more confusing. However again there are still some English mistakes and the paper should be edited for English.

it should be first read by a native English: the paper must be rewritten by correcting mistakes and making sure that certain sentences are no longer confusing. These confusing sentences should be rephrased.

Among authors’ answers, I unfortunately did not find answers to my methodological question: can the authors make sure that the chosen statistical methods were appropriate? Can they argue on that point?

Lines 175-176: the authors mention an increase while the tables evoke a decrease. Is it coherent? Please argue or fix it.

Lines 223-234: is the high rate of hospitalisations only due to under-diagnosis and / or under reporting? Can this be due to a better patients’ care”?

In addition, there are a number of few minor remarks that can also be made afterwards:

Line 20: there is “4)” at the beginning of this line, but the authors evoked a first and a second outcome; does it correspond to the third? The fourth??

Line 38: “re” should read “is”

Lines 40-41: the main route is contaminated food and secondary person to person: this should be emphasized

Lines 48-49: not clear whether the authors 1/ speak of HIV infection or seroprevalence and 2/ if this implies that there is 70% seroprevalence in Africa or that 70% of seropositive in the world are African (??)

Line 52: “in” should read “at”

Line 55: “found” or “estimated”?

Line 57: please add “products after “poultry”

Line s59-61: are refs nb 9 and 10 appropriate / relevant and do they illustrate the text?

Line 70 there is a missing space in “38 437239”

Line 94: “investigate”

Line 105: sentence is not clear; is really the chief sanitary inspector the “central body of government administration”?

Lines 114-115: why do the authors use previous currency and not only Euros?

Line 120 (and elsewhere in the text lines 191, 192…): “Mann–Whitney U test”

Lines 120-121: please fix the change in font height

Line 121: “declared” should read “considered”

Line 146: “during” should read “over”

Line 174: “decrease” should read “decreased”

Line 202: EPSA or EFSA?

Line 216: increased or increases?

Line 221: superfluous space before “suggests”; “of faster” should read “a faster”

Line 225: missing space before “In”

Lines 244-148: please rephrase the sentence in parentheses (or remove it?)…

Line 276: “obliged” should read “mandatory”

Line 279 please italicise “Salmonella”

List of references:

Please italicise “Salmonella in titles

Please remove unnecessary capital letters  in the titles

Author Response

Dear Reviewer,

Thank you for your suggestions and mistakes you found in the text. We implemented all of them and corrected the paper. Here you can find all corrections and answers for your questions.

The paper is still interesting, and should be considered for publication. Some confusing sentences have been rephrased and are no more confusing. However again there are still some English mistakes and the paper should be edited for English.

it should be first read by a native English: the paper must be rewritten by correcting mistakes and making sure that certain sentences are no longer confusing. These confusing sentences should be rephrased.

We decided to send our manuscript to the English editing office. We are waiting for a response.

Among authors’ answers, I unfortunately did not find answers to my methodological questioncan the authors make sure that the chosen statistical methods were appropriate? Can they argue on that point?

Dear Reviewer, we are very sorry for the missing answer of the question about choice of the statistical methods. In the chapter Materials and Methods on the page number 5 in the Manuscript we described it. 

All explored parameters were not normally distributed. It was confirmed by Kolmogorov-Smirnov test. 

Therefore, for comparative analysis of the means the non-parametric tests were used: for paired samples Wilcoxon test and for unpaired samples Mann Whitney U test. The correlation analysis between parameters was done with Spearman test. A p value < 0.05 was declared as statistically significant. Because all explored parameters (without exception) were not normally distributed, we described the choice of statistical tests just in Methods, without a recurrence in chapter Results.

We would be glad to have your suggestions if the statistical tests selection was not appropriate.

Lines 175-176: the authors mention an increase while the tables evoke a decrease. Is it coherent? Please argue or fix it.

We would like to explain it. Firstly, we talked about the absolute value of hospitalisations which mainly decreased as the number of infections at all. Then we discussed about the relative value because compering number of hospitalisations to the number of infections we could observe increasing percentage in the last study period. In the table number 1 are presented mean absolute values so we could describe it as a decreasing value. So we corrected it in the paper to make to more clear.

Lines 223-234: is the high rate of hospitalisations only due to under-diagnosis and / or under reporting? Can this be due to a better patients’ care”?

Thank you for that question. We discussed this problem with an expert in infectious diseases who works at hospital for the last 20 years and of course fortunately we have a better medical care system, and more medications are easily available nowadays, but it is not clear if we could connect it to the higher number of hospitalisations. It does not seem to be a direct reason of higher hospitalisation percentage because the absolute value of hospitalisation is lower. We added in the paper a sentence about this. “Certainly, after 2004, the quality of diagnostics and therapy improved in the Polish healthcare system. Nevertheless, the increase in the ratio of NTS hospitalization to the prevalence in according to the development in healthcare system remains unclear, because the NTS hospitalization recommendations have not changed.”

In addition, there are a number of few minor remarks that can also be made afterwards:

Line 20: there is “4)” at the beginning of this line, but the authors evoked a first and a second outcome; does it correspond to the third? The fourth??

It was a misspelling. We removed it.

Line 38: “re” should read “is”

We corrected it.

Lines 40-41: the main route is contaminated food and secondary person to person: this should be emphasized

We changed the order of listed routes and that makes it more clear what is more important than other.

Lines 48-49: not clear whether the authors 1/ speak of HIV infection or seroprevalence and 2/ if this implies that there is 70% seroprevalence in Africa or that 70% of seropositive in the world are African (??)

1/ We were meant infections. 2/ We corrected the sentence by adding: over 70% of HIV infections are the part of global burden

Line 52: “in” should read “at”

We corrected it.

Line 55: “found” or “estimated”?

We corrected it for “estimated”.

Line 57: please add “products after “poultry”

We corrected it.

Line s59-61: are refs nb 9 and 10 appropriate / relevant and do they illustrate the text?

We corrected the order of references.

Line 70 there is a missing space in “38 437239”

We corrected it to 38 437 239.

Line 94: “investigate”

We corrected it.

Line 105: sentence is not clear; is really the chief sanitary inspector the “central body of government administration”?

Thank you for that indication. State Sanitary Inspectorate Inspection is subordinate to the Ministry of Health and presided by the Chief Sanitary Inspector. We clarified it in the paper. 

Lines 114-115: why do the authors use previous currency and not only Euros?

We decided to present values in Polish currency too. The present currency in our country is still PLN.

Line 120 (and elsewhere in the text lines 191, 192…): “Mann–Whitney U test”

We corrected it.

Lines 120-121: please fix the change in font height

We corrected it.

Line 121: “declared” should read “considered”

We corrected it.

Line 146: “during” should read “over”

We corrected it.

Line 174: “decrease” should read “decreased”

We corrected it.

Line 202: EPSA or EFSA?

It supposed to be EFSA. We corrected it.

Line 216: increased or increases?

It supposed to be “increases”. We corrected it.

Line 221: superfluous space before “suggests”; “of faster” should read “a faster”

We corrected it.

Line 225: missing space before “In”

We corrected it.

Lines 244-148: please rephrase the sentence in parentheses (or remove it?)…

We removed a part of that sentence.

Line 276: “obliged” should read “mandatory”

We corrected it.

Line 279 please italicise “Salmonella”

We corrected it.

List of references:

Please italicise “Salmonella in titles

We corrected it.

Please remove unnecessary capital letters in the titles

We corrected it.